# NAText: Faster Scene Text Recognition with Non Autoregressive Transformer

## Abstract

Autoregressive-based attention methods have made a significant advance in scene text recognition. However, the inference speed of these methods is limited due to their iterative decoding scheme. In contrast, the non-autoregressive methods adopt the parallel decoding paradigm, making them much faster than the autoregressive decoder. The dilemma is that, though the speed is increased, the non-autoregressive methods are based on the character-wise independent assumption, making them perform much worse than the autoregressive methods. In this paper, we propose a simple non-autoregressive transformer-based text recognizer named NAText , by proposing a progressive learning approach to force the network to focus on hard samples and learn the relationship between characters. Furthermore, we redesign the query composition by introducing positional encoding of the character center. And it has more clear physical meanings than the conventional one. Experiments show that our NAText helps to better utilize the positional information for 2D feature aggregation. With all these techniques, the NAText has achieved competitive performance to the state-of-the-art methods. The code will be released.

## 1 Introduction

Reading and processing text from natural scenes has a lot of applications in reality, such as reading road signs, billboards, product labels, logos, etc. Due to its high-demanding characteristics, scene text recognition has attracted a lot of researchers and has been studied for years. Recently, autoregressive methods have achieved great success in scene text recognitionYue et al. (2020)Li et al. (2019)Zhong et al. (2022)Lee et al. (2020b). Structurally, they usually consist of an encoder to extract image features and an autoregressive decoder to transcribe the encoded features into text sequence. By the attention mechanism and autoregressive decoding style, the autoregressive models can extract robust and discriminative features for scene text.

Although the autoregressive models have many advantages in recognition accuracy, the employment of the iterative decoding style results in extremely low efficiency, especially for long text. In contrast, the non-autoregressive models adopt a parallel decoding paradigm. They share similar decoder structure with their autoregressive counterparts but run much faster. As there is no free lunch, while increasing the speed, the performance suffers greatly. For example, in machine translation, the naive non-autoregressive model performs 4% lower than autoregressive modelsGu et al. (2018). In scene text recognition, we notice that in some recent workQiao et al. (2021)Bautista & Atienza (2022) the non-autoregressive recognizers perform about 2% lower than autoregressive models. This is consistent with our experimental findings that the non-autoregressive model performs 1.7% lower in regular text and 3.4% lower in irregular text. For scene text recognition, such performance drop is considerable. Despite the non-satisfactory performance, the huge advantage in decoding speed is too attracting that some of the most recent workYu et al. (2020a)Fang et al. (2021a)Qiao et al. (2021) on scene text recognition still attempts to adopt such parallel decoding scheme. To remedy the performance degeneration, they either introduce large language modelsFang et al. (2021a) to correct the error prediction in a post-process manner or design a heavy predictionQiao et al. (2021) pipeline. These methods are all designed to be very complex and require considerable computational burden. In a sense, they do not fundamentally solve the problem of why non-autoregressive models get inferior performance. Therefore, in this case, we try to answer the question: Is it possible to design

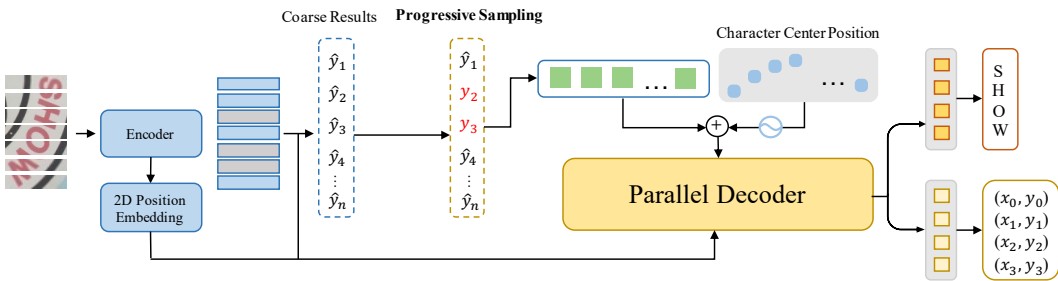

Figure 1: Schematic overview of two stage structure of NAText. Note that the progressive sampling is only applied during training.

a non-autoregressive scene text recognizer to match its autoregressive counterpart in performance without resorting to other language models or any complex decoding pipeline?

In this paper, we propose NAText as a solution to the above question. NAText is short for Non-Autoregressive scene Text recognizer. It uses a simple encoder-decoder structure without extra modules and extra post-process. We start by digging into the inferior performance and find that the harder situations usually suffer more significant drop, e.g., the irregular text(Table-6 and the longer text(Figure-3a. To better resist the performance drop in these harder situations, we propose three techniques. First, we argue that the independent assumption adopted by the non-autoregressive model is the main reason to blame. For hard cases, the character-wise inter-dependency provides rich information for prediction. We drop the independent assumption entirely and design incremental learning to enforce mutual constraints on character predictions. Specifically, during training, we sample some characters and replace them with their ground truth token embeddings, and force the remaining characters to be learned under this condition. In this way, the network will gradually capture the character-wise relationships. Second, we design progressive sampling to force the training to focus on hard characters. During sampling, the confident predictions are more likely to be replaced, leaving the hard characters to be learned. Together with the first technique, we name this learning scheme the progressive sampled learning. Third, to better capture each character's visual information, we follow the recently proposed DAB-DETRLiu et al. (2022) to adopt a re-designed decoder structure in which the character center explicitly models the positional information. It unifies the physical meaning of the positional encoding from the image features and query embeddings. This is in contrast to the inconsistency of the positional encoding of query and encoded features of traditional decoders.

We experiment on six popular scene text recognition benchmarks to verify the effectiveness of NA-Text. Detailed exploration into each part is also conducted. In summary, this paper's contributions mainly include: 1) We propose NAText as a simple and powerful non-autoregressive scene text recognizer. It is both fast and strong compared to most recent work. 2) We research deep into the reason behind the inferior performance of non-autoregressive decoding and propose progressive sampled learning to overcome it. 3) We re-design the decoder structure to utilize the positional information that leads to better visual perception.

## 2 RELATED WORK

Based on the topic of our method, we roughly divide the current methods into autoregressive and non-autoregressive methods.

**Autoregressive Text Recognition**. Autoregressive methods can be grouped into 1D-attention based and 2D-attention based. Earlier methods usually encode the image features to 1D feature sequence and use 1D attention in the decoding period. For example, the $R^2$AMLee & Osindero (2016) design an autoregressive CNN that can capture broader features as the feature extractor and a 1D-attention-based decoder to transcribe the sequence. FANCheng et al. (2017) employs a focusing attention mechanism to automatically draw back the attention drift. Fang et al. (2018) proposes a fully CNN-based network to extract visual and language features separately. However, these methods usually lack the ability to process irregular text(e.g., curved, rotated). To this end, recent methodsLee et al.

(2020b)Fang et al. (2021a)Bautista & Atienza (2022)Qiao et al. (2021) of scene text recognition usually encode the image into 2D features and adopt the 2D attention in the decoder. With the help of 2D attention, they consistently show strong performance on irregular text recognition. In this paper, we also choose the 2D attention-based transformer to build our baseline method. We mainly focus on the design of the decoder query and show that by re-designing the query, the simple and concise structure can also lead to powerful performance.

**Non-Autoregressive Text Recognition**. Non-autoregressive methods predict the target sequence at a single iteration or constant time independent of the sequence length. They can be categorized into three groups: the CTC-Based methods, the segmentation-based methods, and the attention-based. The attention-based non-autoregressive methods have been widely applied in machine translationGhazvininejad et al. (2019)Gu et al. (2018)Wang et al. (2019)Qian et al. (2021), auto speech recognitionTian et al. (2020)Chi et al. (2021)Chan et al. (2020) and capture generationGuo et al. (2021). In comparison, there is less workQiao et al. (2021)Fang et al. (2021a)Yu et al. (2020a) for the research of non-autoregressive model on scene text recognition. Recent methodsFang et al. (2021a)Yu et al. (2020a) on scene text recognition that is relative to the non-autoregressive model mainly pay attention to the employment of language models to assist the text recognition. They usually design a complex multi-model system to get a high-performance text recognizer, but the efficiency of the model is often overlooked. In contrast, our work is focused on the nature of the non-autoregressive model itself. This work aims to explore how we can design a simple and powerful non-autoregressive model that keeps the merit of high efficiency and high performance.

**Masking Technique**. The masking technique has been widely applied to the pre-training of transformersDevlin et al. (2018)Joshi et al. (2020)Song et al. (2019)Lewis et al. (2019)Song et al. (2020). Different from these works, the masked tokens in NAText are replaced with their ground truth embeddings. They are ignored in loss calculation. For the masking technique, the most relevant works are Mask-PredictGhazvininejad et al. (2019) and GLMQian et al. (2021) proposed for machine translation. They both randomly mask tokens to replace and the remaining tokens are predicted under such condition. However, the random sampling that is suitable for machine translation has little effect on scene text recognition. The reason is that the task of scene text recognition has to deal with lots of noised inputs, e.g., blurry, occluded, and incomplete while the inputs of machine translation are clean. Based on the task characteristics, we follow the idea of hard sample mining and propose progressive sampling to feed more informative samples for training, which is proved crucial for scene text recognition. Besides, they need multiple decoding times, either for training or testing. While we design a two-stage decoding scheme to avoid repetitive decoding. In all, the masking technique in NAText is specially designed for scene text recognition. It is both concise and effective.

## 3 PROPOSED METHOD

### 3.1 OVERALL ARCHITECTURE

The structure of NAText is depicted in Figure 1. The NAText adopts the transformer based encoder-decoder structure. Given an input image, the encoder will extract the image features and generates the coarse sequence prediction. Then the predicted sequence is fed to the decoder to generate the final refined result. Along with the sequence output, the character coordinate will be predicted by the regression head.

NAText mainly optimizes the decoder structure. Compared to conventional text recognizers, we highlight two differences in structure. The first is the parallel decoding style. Parallel decoding does not need much modification to the decoder structure. It only needs to discard the masking operation used to guarantee uni-directional self-attention. The second is that we introduce the concept of location query into the decoder structure. It has the exact physical meaning and makes the decoding process easier to interpret while also performing better.

**Query Composition.** Inspired by the recent advance in object detection, we follow DAB-DETRLiu et al. (2022) to introduce the positional query(embedding) into the decoding process. For clarity, we refer to the original query embedding in conventional decoder as content embedding, denoted as $c_q$. In self-attention(Shown in Figure-2), the query, key, and value embeddings are obtained by

$$\text{query}, \text{key} := c_q + p_q + s_q, p_q = \text{PE}(x, y) \qquad \text{value} := c_q \qquad (1)$$

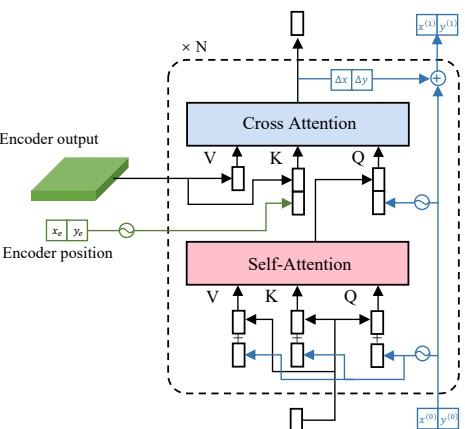

Figure 2: Decoder structure of NAText.

where $c_q$ and $s_q$ denote the content embedding and the sequential positional encoding used in conventional decoders respectively. $p_q$ denotes the newly introduced positional embedding. $(x, y)$ denotes character's center coordinate. PE is the positional encoding function. Following previous work, we use the sinusoidal function to generate the positional encoding.

Note that in conventional decoder, the query and key for self-attention are calculated by $c_q + s_q$.

## 3.2 DECOUPLED NON-AUTOREGRESSIVE DECODER

The detailed structure of the decoder is shown in Figure-2. In cross attention, the query, key, and value embeddings are defined by

$$\text{query} := \text{CAT}(c_q, p_q) \qquad \text{key} := \text{CAT}(X, X_p) \qquad \text{value} := X \tag{2}$$

where CAT is the concatenation operation. $X$ denotes the encoded image features. $X_p$ denotes the per-pixel positional encoding of $X$. The encoding function of $p_q$ and $X_p$ is the same.

Note that in conventional decoder, the query for cross attention contains only $c_q$. The key is obtained by $X + X_p$.

Based on the query design, the cross attention is decomposed into content attention and spatial attention. Given the query $q$, $k$, $v$, the cross attention of decoder can be formulated as:

$$\text{Attention}(q, k, v) = \text{softmax}(\frac{qk^T}{\sqrt{d_k}}v), \tag{3}$$

where the $d_k$ is the channel dimension, and the attention part of $qk^T$ can be decomposed into the two dot-products of content embeddings and positional embeddings respectively $c_q^T X + p_q^T X_p$. Thus, the cross attention can be viewed as the feature aggregation process influenced by both the content information and spatial information.

**Coordinate Regression.** Unlike the traditional text recognition model, we design the decoder output to include both character categorization and coordinate regression. The character's coordinate is regressed via an iterative style. Given the coordinate prediction from previous decoder layer $(x', y')$, the current coordinate prediction is calculated by

$$(x, y) = \sigma(\text{FFN}(f) + \sigma^{-1}(x', y')), \tag{4}$$

where $\sigma$ is the sigmoid function used to normalize the coordinates to range (0,1) and $\sigma^{-1}$ is the reverse sigmoid function. FFN aims to regress the relative offset from the decoder embedding $f$.

## 3.3 PROGRESSIVE SAMPLED LEARNING

In this part, we start by comparing the different probability models between autoregressive and non-autoregressive methods. It partially explains the reason for the non-autoregressive model's inferior performance. Then, we introduce the progressive learning strategy for non-autoregressive models.

**Assumptions behind autoregressive and Non-autoregressive models**. The text recognition can be formally defined as a sequence generation problem: given the source features $X$ extracted from the image, to generate the target character sequence $Y = \{y_1, y_2, ..., y_T\}$ according to the conditional probability $P(Y|X; \theta)$, where $\theta$ is the parameter set of the model. For autoregressive models, the conditional probability is factorized to maximize the following likelihood:

$$L_{rec} = \log P(Y|X; \theta) = \sum_{t=1}^{T} \log p(y_t|y < t, X; \theta), \tag{5}$$

where $y < t$ is the short for $\{y_1, ..., y_{t-1}\}$. The autoregressive factorization adopts the assumption of an uni-directional inter-dependency between characters where each token is conditioned by the previous token sequence.

For non-autoregressive models, each character is assumed to be independent for parallel decoding. The independent factorization is written as

$$L_{nrec} = \sum_{t=1}^{T} \log P(y_t|X; \theta). \tag{6}$$

The autoregressive factorization in Eq-5 and non-autoregressive factorization in Eq-6 both serve as the approximation to the conditional probability $P(Y|X; \theta)$. As the independent assumption does not hold in general, the corresponding factorization deviates further from the real conditional probability $P(Y|X; \theta)$. So non-autoregressive models trained under such biased objective gets inferior performance.

**Rectified Learning Objective**. Based on the above analysis of the two factorizations Eq-5, Eq-6, we argue that the independent assumption should be abandoned and more suitable factorization needs to be designed to better fit the real optimization objective. In our design, the character-wise dependency is also encouraged. Different from the autoregressive factorization, we encourage the model to learn dependency from any other characters in the sequence, not just the previous characters. Specifically, we design the following factorization:

$$L_{PM} = \sum_{y_t \notin \mathbb{PS}(Y, \hat{Y}),} \log p(y_t|\mathbb{PS}(Y, \hat{Y}), X; \theta), \tag{7}$$

where $Y$ is the ground truth sequence and $\hat{Y}$ is the predicted sequence. $\mathbb{PS}(Y, \hat{Y})$ denotes the sampling operation based on the ground truth and predicted sequence. The sampled result is a subset of tokens of $Y$ that will be directly replaced with the corresponding character embedding, serving as the prior knowledge input to the decoding process. For example, given $Y = \{y_1, y_2, y_3, y_4, y_5\}$ and $\mathbb{PS}(Y) = \{y_2, y_3\}$, the input queries corresponding to $\{y_2, y_3\}$ will be replaced by their target character embeddings, which are obtained from the softmax embedding matrix. The sampled tokens will not be considered during the loss calculation. Only the remaining $\{y_1, y_4, y_5\}$ will contribute to the final loss. In this way, the learning objective is to learn a refinement model $\theta$ that can predict the remaining tokens given the ground truth of the sampled tokens and source image features $X$.

**Progressive sampling**. Following the designed factorization in Eq-7, we find that the naive random sampling even leads to worse performance. The reason is twofold. First, as most characters are easy samples, the training hardly focuses on the informative samples. Such scheme is inefficient and leads to low performance. Second, the model is trained and tested under different conditions. During training, the model is always encouraged to predict with the help of extra knowledge while during testing, there is not. In other words, the model is tested in a more difficult condition than training. Therefore, we design progressive sampling scheme in which the characters are sampled based on their predicted confidence. Specifically, given the predicted sequence $\hat{Y}$, the ground truth sequence $Y$ and the confidence $C = \{c_1, c_2, ..., c_T\}$ corresponding to $\hat{Y}$, we first determine the sampling number by $N = \lambda \cdot \sum_{t=1}^{T}(c_t < \tau)$, where $\tau$ is the confidence threshold and $\lambda$ is the hyper-parameter controlling the sampling ratio. Then the top-N confident characters are sampled as the prior knowledge, forcing the network to learn the remaining hard samples.

The designed progressive sampling can well solve the above problems. First, the remaining characters are unconfident or even incorrect predictions. They are more informative for the training

process. Second, during training, as the overall predictions become more and more confident, the sampling number will gradually reduce. At the beginning phase of training, the model is encouraged to learn under extra knowledge. While at the end phase, the model is forced to learn to predict in parallel. This is in accord with our expectations for progressive learning, by which the learning difficulty gradually increases.

## 3.4 OPTIMIZATION OBJECTIVE

We further design a two-stage decoding scheme to simplify the progressive sampled learning. The extra bonus is that the first stage decoding will also bring performance improvement.

To get the characters' predicted confidence for progressive sampling, the direct thought is a two-pass decoding manner. In the first pass, the confidence is generated. In the second pass, the progressive sampling is applied. We argue that this way is tedious and unnecessary. In our two-stage decoding scheme, we directly use the encoder network to make a coarse sequence prediction, which serves as the predicted confidence for the progressive sampling. The coarse prediction can also serve as the prior knowledge of the decoding process. The pipeline of the two-stage proceeds as follows, given the image, features output from the encoder $X$, and the target sequence $Y$, we first use a prediction head FFN to get the per-pixel classification $S = \text{FFN}(X)$. Then we apply the connectionist temporal classification(CTC) loss as the supervision between the predicted logits $S$ and target $Y$.

$$\mathcal{L}_{enc} = \text{CTCLoss}(S, Y). \tag{8}$$

The predicted sequence is obtained via evaluating the result of $\text{argmax}(S)$. In this way, the encoder output is capable of making coarse predictions. After we get the coarse sequence prediction $\hat{Y}'$, we take the corresponding character embedding $\{h_1, h_2, ..., h_{T'}\}$ from the decoder softmax matrix to substitute the content part of the initial query. Here, $T'$ denotes the prediction length of the coarse sequence.

Using the per-pixel classification $S$, we can also get the coarse normalized character coordinate for each predicted token. The coordinate is used to generate the initial positional embedding of the query. Finally, using Eq-2, we combine the two parts to get the initial query proposals. Considering that the coarse-predicted sequence length $T'$ may be incorrect, the number of queries of the decoder remains unchanged. We only use the proposals to replace the first $T'$ queries.

The training objective includes the CTC loss applied to the encoder output and the character classification and regression loss applied to decoder output. We use the cross entropy loss for classification and the L1 loss for regression. For decoder, the loss is formulated as

$$\mathcal{L}_{dec} = \frac{1}{L} \sum_{l=1}^{L} \sum_{t=1}^{T} (\log \hat{p}_l(y_t) + \mathbb{1}(y_t \neq [EOS]) \mathcal{L}_{reg}(\hat{c}_{lt}, c_t)), \tag{9}$$

where $\hat{p}_l(y_t)$ denotes the predicted probability corresponding to ground truth token $y_t$ by the $l$-th decoder layer. $\hat{c}_{lt}$ and $c_t$ represents the $l$-th predicted and ground truth character coordinates, respectively. They are both normalized by the image scale. We note that the character coordinate is not always annotated. Usually, the synthetic datasets contain this annotation, while the real dataset does not. Thus, the regression loss is only applied when the annotation is available. The final loss is the weighted sum of the Eq-8 and Eq-9.

## 4 EXPERIMENTS

### 4.1 IMPLEMENTATION DETAILS

**Structure** We follow SATRNLee et al. (2020b) to build the basic model structure. Specifically, the number of hidden units for self-attention layers is 512. The numbers of self-attention layers in the encoder and decoder are $N_e = 12$ and $N_d = 6$ respectively. We set the number of classes to 91, including 10 digits, 52 case-sensitive letters, 28 punctuation characters, and an $< EOS >$ token. Specially, similar to a left-to-right autoregressive decoder, $< EOS >$ token is viewed as the end of the sequence, so there is no need to predict the sequence length in advance.

**Optimization.** All experiments are conducted on servers with 8 NVIDIA Tesla A100 GPUs. For fair comparison, all models are trained from scratch using Adam optimizer. The whole training

Table 1: Accuracy comparison with other methods. Note that ABINet-LV-NL refers to ABINet-LV without language model.

| Decoder Type | Method | Regular Text | | | Irregular Text | | | Avg |
| --- | --- | --- | --- | --- | --- | --- | --- | --- |
| | | III5K | SVT | IC13 | IC15 | SVTP | CT80 | |
| AR | SPDNChen et al. (2022) | 94.1 | 89.9 | 91.7 | 77.9 | 79.8 | 81.6 | 85.8 |
| | DANWang et al. (2020) | 94.3 | 89.2 | 93.9 | 74.5 | 80.0 | 84.4 | 86.1 |
| | RobustScannerYue et al. (2020) | 95.3 | 88.1 | 94.8 | 79.5 | 77.1 | 90.3 | 88.6 |
| | SGBANetZhong et al. (2022) | 95.4 | 89.1 | 95.1 | 78.4 | 83.1 | 83.1 | 88.7 |
| | SARLi et al. (2019) | 95.0 | 91.2 | 94.0 | 78.8 | 86.4 | 89.6 | 89.2 |
| | TREFEZhang et al. (2022) | 94.8 | 91.3 | 95.4 | 84.0 | 84.5 | - | - |
| | SATRN(Reproduced)Lee et al. (2020a) | 95.9 | 93.4 | 96.4 | 83.1 | 88.6 | 89.2 | 91.4 |
| | PARSeqABautista & Atienza (2022) | 97.0 | 93.6 | 96.2 | 82.9 | 88.9 | 92.2 | 91.9 |
| NAR+LM | ABINet-LV(Reproduced)Fang et al. (2021b) | 95.3 | 93.4 | 95.0 | 79.1 | 87.1 | 89.7 | 89.8 |
| | SRNYu et al. (2020b) | 94.8 | 91.5 | 95.5 | 82.7 | 85.1 | 87.8 | 90.2 |
| NAR | CRNNShi et al. (2016) | 78.2 | 80.8 | 86.7 | - | - | - | - |
| | ViTSTRAtienza (2021) | 88.4 | 87.7 | 92.4 | 72.6 | 81.8 | 81.3 | 83.8 |
| | SRN w/o GSTMYu et al. (2020b) | 92.3 | 88.1 | 93.2 | 77.5 | 79.4 | 84.7 | 86.7 |
| | SATRN-NAR | 93.8 | 90.0 | 95.4 | 78.8 | 86.1 | 84.5 | 88.6 |
| | ABINet-LV-NLFang et al. (2021b) | 94.6 | 90.4 | 94.9 | 81.7 | 84.2 | 86.5 | 89.6 |
| | PARSeqNBautista & Atienza (2022) | 95.7 | 92.6 | 95.5 | 81.4 | 87.9 | 91.4 | 90.7 |
| | NAText W/O PO | 95.6 | 93.0 | 96.1 | 82.1 | 86.5 | 91.3 | 90.9 |
| | NAText | 95.8 | 93.4 | 96.3 | 82.4 | 86.7 | 90.3 | 91.1 |

process contains 6 epochs, and the initial learning rate is $3 \times 10^{-4}$ while decreases to $3 \times 10^{-5}$ at the $3^{rd}$ epoch and $3 \times 10^{-6}$ at the $5^{rd}$ epoch. The batch size is set to 256.

## 4.2 DATASETS

We use two publicly available synthetic datasets, i.e., Mjsynth(MJ)Jaderberg et al. (2014), and SynthText(ST)Gupta et al. (2016) as training datasets and test on six standard benchmarks: IIIT 5k-word (IIIT5K) Mishra et al. (2012), CUTE80 (CUTE) Risnumawan et al. (2014), Street View Text (SVT) Wang et al. (2011), SVT-Perspective (SVTP)Phan et al. (2013), ICDAR 2013 (IC13) Karatzas et al. (2013), and ICDAR 2015 (IC15) Karatzas et al. (2015).

IIIT5K is a large natural scene dataset collected from Google, containing 5000. CUTE contains 288 cropped high-resolution images, many of which are curved or irregular text images. SVT is a Google Street View dataset, which consists of 647 patches for testing. SVTP consists of 639 patches, cropped from side view snapshots in Google Street View. In SVTP, many patches encounter severe perspective distortions. IC13 contains 848 patches for training and 1095 for evaluation. IC15 consists of incidental scene text images under arbitrary angles. Therefore, most word patches in this dataset are irregular (oriented, perspective, or even curved).

## 4.3 PERFORMANCE COMPARISON

**Comparison to State of the Art.** We compare NAText to three types of methods including the autoregressive models(AR), the pure non-autoregressive models(NAR) and the language model enhanced non-autoregressive models(NAR+LM). The results are shown in Table-1. For the data processing, we strictly follow the setup of PARSeqABautista & Atienza (2022). Specifically, images are resized to $32 \times 128$ with data augmentation such as geometric transformation, image quality deterioration and color jitter, etc. We reproduce SATRN and its non-autoregressive version to serve as the baseline methods. It is seen that the NAR version of SATRN performs 2.8% lower than its AR version. For fair comparison, we also report the result of NAText without positional supervision(NAText W/O PO) in Table-1. The proposed NAText performs best among all the current non-autoregressive models. The comparison to autoregressive models and language model enhanced non-autoregressive based models is also challenging. Specifically, compared with its baseline method, NAText increases the overall performance by 2.5%. Besides, it also outperforms some language model based methods. These comparison results validate the effectiveness of NAText.

**Speed Comparison.** For fair speed and accuracy comparison, we re-implement the SATRN and its non-autoregressive version . The speed and accuracy comparison is shown in Table-6. NAText almost shares a similar structure with the naive non-autoregressive version. They are almost three times faster than the autoregressive model on average. But the SATRN-NAR suffers a significant performance drop on both regular and irregular text. NAText nearly matches the SATRN with litter speed decrease compared with SATRN-NAR.

**Contribution of Each Part.** We experiment to figure out the contribution of each part, namely, PO for positional query design, PS for progressive sampling, and TS for the two-stage scheme. Note that for the setting of PS, as we can not get the confidence from the first stage predictor, the confidence is generated via the decoder. The decoder runs twice during training. The results are shown in Table-3. We can see that each part will effectively improve the baseline performance. While when all modules are applied, the whole performance will further be increased. Specifically, NAText will improve by 1.6% in the regular text and 2.8% in the irregular text.

**Comparison under Different Text Length.** In Figure-3a, we compare NAText with the autoregressive and non-autoregressive baseline under different text length. We can see that the SATRN-NAR performs especially poorly for the long text. It is lower by 3% than its AR version when the text length is greater than 10. Our NAText performs better than SATRN for the short text and medium-length text. Although the performance for long text is still inferior to the autoregressive model(-1.3%), the performance gap under such setting has been improved by 1.7%.

Table 2: Comparison between fixed-ratio sampling and progressive sampling.

| | ratio | Accuracy | |
|---|---|---|---|
| | | Regular | Iregular |
| Fix ratio | 0.00 | 93.3 | 80.5 |
| | 0.25 | 93.9 | 80.4 |
| | 0.50 | 93.6 | 80.4 |
| | 0.75 | 93.5 | 80.3 |
| | 1.00 | 93.3 | 79.9 |
| Progressive | - | **94.0** | **81.3** |

Table 3: Ablation on the effect of each module. **PO** is for positional query design. **PS** is for progressive sampling. **TS** is for two-stage training and testing.

| Module | | | Accuracy | |
|---|---|---|---|---|
| PO | PS | TS | Regular | Irregular |
| | | | 93.3 | 80.5 |
| ✓ | | | 93.8 | 81.3 |
| | ✓ | | 94.0 | 81.3 |
| | | ✓ | 93.9 | 81.8 |
| ✓ | ✓ | ✓ | **94.9** | **83.3** |

## 4.4 ABLATION STUDY

For fair comparison, no augmentation is used for experiments in this part.

**Query Design.** We experiment with different designs of queries to demonstrate the effectiveness of NAText. The query design mainly influences the cross attention in the decoding process. NAText uses the concatenation of content embedding and positional embedding to form the query embedding. We denote it as $\text{CAT}(c_q, p_q)$. Note that the positional embedding for NAText has a clear physical explanation. It is the encoding of the characters' 2D coordinate. Such positional embedding has never been adopted in the conventional recognizer. We choose several types of designs for comparison. (1) $c_q$:The query contains only the content part. The cross attention is obtained by computing the dot product of the projection of query content embedding $c_q$ and image features $X$. (2) $p_q$: the query contains only the positional part. The cross attention is obtained by computing the dot product of query positional embedding $p_q$ and image positional embedding $X_p$. (3) $\text{ADD}(c_q, p_q)$: The query is the summation of the content embedding and the positional embedding. The conventional recognizer usually uses the $c_q$-only for the query. It does not mean that they directly drop the positional information. Rather, the content query will still be added by a sinusoidal encoding that represents the sequential order. The results are shown in Table-4. We can see that only using the positional embedding performs the worst, both in the regular and irregular text. The conventional style $c_q$-only will be further improved when the positional information is introduced. And we find that the concatenation performs better than the add operation. It is consistent with the conclusion in previous workLiu et al. (2022).

To further understand the influence of content embedding and positional embedding, we conduct a quality experiment by visualizing the attention plot of each component. It is shown in Figure-

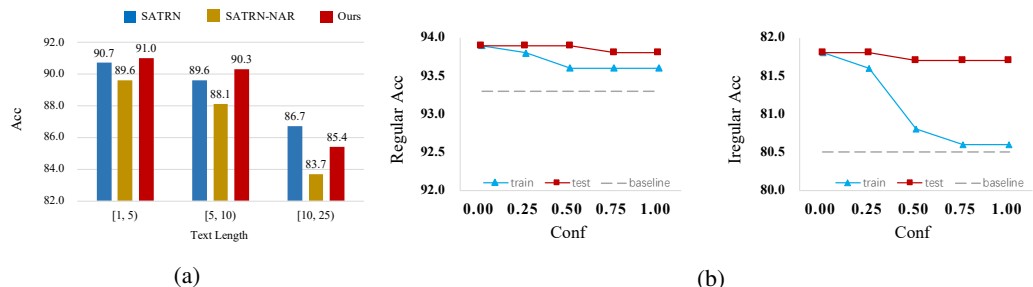

(a)                                                                                  (b)

Figure 3: (a)Accuracy of different lengths on test set.(b)Ablation on the two-stage confidence threshold on regular and irregular text. Conf is short for confidence threshold. The influence on both train and test phase are reported.

Figure 4: Quality visualization of attention plot. For our query design, the final attention(**3-rd** coloum) is the composition of the position part(**1-st** coloum) and content part(**2-nd** coloum). For conventional query design. the attention(**4-th** coloum) only contains the content part.

4. We find the effect of the two parts has much difference. The content embedding will attend to many of the neighboring characters while the positional embedding will strictly focus on the current character. Thus, with the help of the positional part, the final attention of NAText is more accurate than the naive non-autoregressive model.

**Influence of the Positional Supervision.** We have validated the effectiveness of the query design. While we have no idea whether the contribution comes from the query design or the positional supervision. So in this experiment, we explore the influence of positional supervision on different query compositions. The results are shown in Table-5. We can see that the positional supervision can always improve the performance no matter the query composition. However, for conventional query design that only contains the content part. The improvement is very small. While adding the positional part will further improve the performance by a large margin, +0.4% in regular text and +0.5% in irregular text. We also find that even when no positional supervision is applied, our query design also beat the baseline method. It even surpasses the conventional query design with positional supervision.

**Sampling matters** We explore other alternative sampling strategies for progressive learning(Shown in Table-7. Including the default sampling strategy, we compare five sampling strategies. (1) $c_t$: Each character is sampled under the probability proportional to its predicted confidence $c_t$. (2) $1 - c_t$: Each character is sampled under the probability proportional to its inverted predicted confidence $1 - c_t$. (3) **T-N**: Top-$N$ confidently predicted characters are sampled for replacement. (4) **B-N**: Top-

Table 4: Ablation on the query design: $c_q$ for content embedding only. It is the way that conventional decoder uses. $p_q$ for the positional embedding only. **ADD($c_q, p_q$)** for adding two parts. CAT($c_q, p_q$) for concatenating two parts.

| Design | Accuracy | |
|---|---|---|
| | Regular | Irregular |
| $c_q$ | 93.3 | 80.5 |
| $p_q$ | 93.3 | 80.5 |
| ADD($c_q, p_q$) | 93.5 | 81.3 |
| CAT($c_q, p_q$) | **93.8** | **81.3** |

Table 5: Influence of the positional supervision

| content | position | position supervison | Accuracy | |
|---|---|---|---|---|
| | | | Regular | Irregular |
| ✓ | | | 93.3 | 80.5 |
| ✓ | | ✓ | 93.4 | 80.8 |
| ✓ | ✓ | | 93.6 | 81.0 |
| ✓ | ✓ | ✓ | **93.8** | **81.3** |

Table 6: Speed Comparision

| Method | Accuracy | | FPS |
|---|---|---|---|
| | Regular | Irregular | |
| SATRN | 95.0 | 83.9 | 188 |
| SATRN-NAR | 93.3 | 80.5 | 551 |
| NAText | 94.9 | 83.3 | 543 |

Table 7: Different sampling strategies.

| | SATRN -NAR | rand (GLM) | $c_t$ | 1 - $c_t$ | T-N (ours) | B-N |
|---|---|---|---|---|---|---|
| Regular | 93.3 | **94.0** | 93.6 | 93.7 | **94.0** | 93.4 |
| Irregular | 80.5 | 80.1 | 81.0 | 80.4 | **81.3** | 79.9 |

$N$ un-confident predicted characters are sampled for replacement. (5) **Rand**: $N$ random characters are sampled for replacement. It is the way that GLMQian et al. (2021) adopts. Intuitively, the $c_t$ and T-N encourage the well-learned characters to be replaced, making the training concentrate more on the hard cases. While the $1 - c_t$ and B-$N$ are the opposite. It is seen that for regular text, all sampling methods can improve performance. While for irregular, only $c_t$ and T-$N$ that follow the idea of hard sample mining can improve the performance. The others all perform even worse than the baseline. In Table-2, we further compare the fixed-ratio sampling strategy and the progressive sampling strategy. The fixed-ratio strategy means the sampling number is always proportional to the text length. We can see that without progressive strategy, the performance is damaged, especially for irregular text.

**Why Two Stage Helps.** As shown in Table-3, the two-stage training and testing scheme can effectively improve the baseline by 0.6% in regular text and 1.3% in two-stage text. While the reason behind the improvement is not fully understood. We experiment with different confidence thresholds for the two-stage scheme. The results are shown in Figure-3b. For results of train, it is obtained by varying the thresh and training the network from scratch. For results of test, we use the best trained network to evaluate various thresholds. It is seen that the confidence has different influence on the train and test phase. For training, the low threshold will lead to better performance. Practically, we set threshold to zero. It means that all predictions from the first stage are used to initialize the query no matter the value of confidence. For testing, although the trend is similar, the influence is relatively small. Even when setting the threshold to 1.0, by which the first stage will never generate proposals, the performance is still better than the baseline. Based on the comparison between train and test, we conclude that the two-stage works in two aspects. First, the extra supervision on the first stage encoder benefits the recognizer, especially for irregular text. Second, the query initialization also helps the decoder to perform better.

## 5 CONCLUSION

In this paper, we propose a simple and powerful non-autoregressive text recognizer NAText. It elegantly solves the problem that non-autoregressive model often performs inferior to its counterpart. Specifically, We rectify the basic assumption and design a progressive sampled learning to help non-autoregressive model to perform better. We also introduce positional encoding that has clear physical meaning for better visual perception. Experiments on various datasets verify the effectiveness of our method.

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

# A   APPENDIX

## A.1   TRADE-OFF BETWEEN EFFICIENCY AND ACCURACY

In this part, we present the speed and accuracy of NAText under different encoder and decoder layers. The results are shown in Table-8.

Table 8: Results for accuracy and speed under different number of layers. Enc and Dec denotes the number of encoder layers and decoder layers, respectively. The accuracy is the weighted average of the 6 scene text recognition benchmarks.

| Enc | Dec | Acc | Time(ms) |
| --- | --- | --- | --- |
| 12 | 1 | 90.6 | 17 |
| 12 | 2 | 90.6 | 18 |
| 12 | 3 | 90.9 | 19 |
| 12 | 4 | 90.9 | 21 |
| 12 | 5 | 91.0 | 22 |
| 12 | 6 | 91.1 | 23 |
| 10 | 6 | 91.0 | 21 |
| 8 | 6 | 90.8 | 19 |
| 6 | 6 | 90.7 | 17 |
| 4 | 6 | 89.4 | 15 |
| 2 | 6 | 88.4 | 13 |

## A.2 Further Comparison with Peer Methods

To further validate the effectiveness of NAText, we design VIT-NAText as a light weight fully transformer-based recognizer. It uses a 12-layer VIT as the encoder. The speed and accuracy comparison is presented in Table-9. Note that for fair comparison, we reproduce ParseqN using the identical experimental conditions with NAText. It gets 90.0, lower than its published result. We suspect that it may have something to do with the hardware conditions. We also find that the author of ParseqN has explained the phenomenon of performance fluctuation in their GitHub issues.

Table 9: Further comparison on speed and accuracy. The accuracy is the weighted average of the 6 scene text recognition benchmarks.

| Method | Acc | Time(ms) |
|---|---|---|
| ParseqA | 91.9 | 37 |
| SATRN | 90.6 | 126 |
| ParseqN | 90.7 | 11 |
| ParseqN(Reproduced) | 90.0 | 11 |
| ABINet | 89.8 | 27 |
| NAText | 91.1 | 23 |
| VIT-NAText | 90.8 | 13 |

## A.3 Threshold for Progressive Learning

In Table-10, we explore the influence of the confidence threshold for progressive learning. The threshold affects the number of samples to be replaced. The larger threshold means more sampling number. When setting threshold to 0, no token is sampled (no character will has lower confidence than 0), which is the baseline method. When setting threshold to 1, it means the number of sampling tokens is always equal to the character length. We can see that the two extremes both get poor performance. In default, we set the confidence threshold to 0.5.

Table 10: Threshold for Progressive Learning

| Conf | 0.0 | 0.1 | 0.3 | 0.5 | 0.7 | 0.9 | 1.0 |
|---|---|---|---|---|---|---|---|
| Acc | 88.2 | 88.9 | 88.5 | 89.0 | 88.5 | 88.8 | 88.2 |

## A.4 Coordinate regression

Most of the regressed coordinates are fairly accurate, though the training of coordinate regression is applied on part of the training dataset. For the qualitative analysis, some samples are selected to visualize the coordinate regression, including both correct and incorrect samples. It is shown in Figure-5. We find that the incorrect predictions are more likely to appear in situations like dense located texts, artistic-styled texts and those with complicated background. We take some representative examples for visualization. They are also challenging cases for scene text recognition.

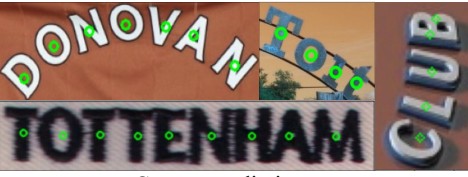

Correct predictions

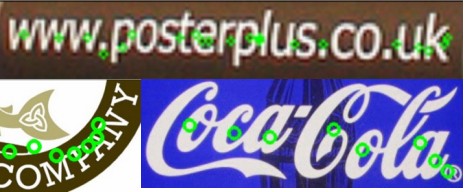

Incorrect predictions

Figure 5: Qualitative results for characters' coordinates regression.

## A.5 Ablation
of Supervision on First Stage

In this part, we further explore into the influence of the supervision applied to the encoder output. In default, we use CTCLoss as the first stage supervision. We change the CTCLoss to cross entropy loss and report the results in Table-11. The CTCLoss will implicitly locate each character in the feature map. While the cross entropy loss simply takes the first K tokens for supervision, where K equals to the word length. We find that the performance is still improved even with cross entropy loss as the supervision. We further

compare the attention map of the encoder output for each character. The quality result is shown in Figure-6. We can see that the attention map with CTCLoss is more accurate and concentrate than the other two. The phenomenon suggests the improvement of the first stage supervision comes from two folds. The first is that the supervision helps the representation of different characters to be more discriminative. The second is that the features of each token in the encoded feature map become more concentrated.

Table 11: Ablation on the supervision for the first stage. CTC is for CTCLoss. CE is for cross entropy loss.

|  | Accuracy | |
| --- | --- | --- |
|  | Regular | Irregular |
| CTC | 0.942 | 0.816 |
| CE | 0.937 | 0.815 |
| Baseline | 0.933 | 0.805 |

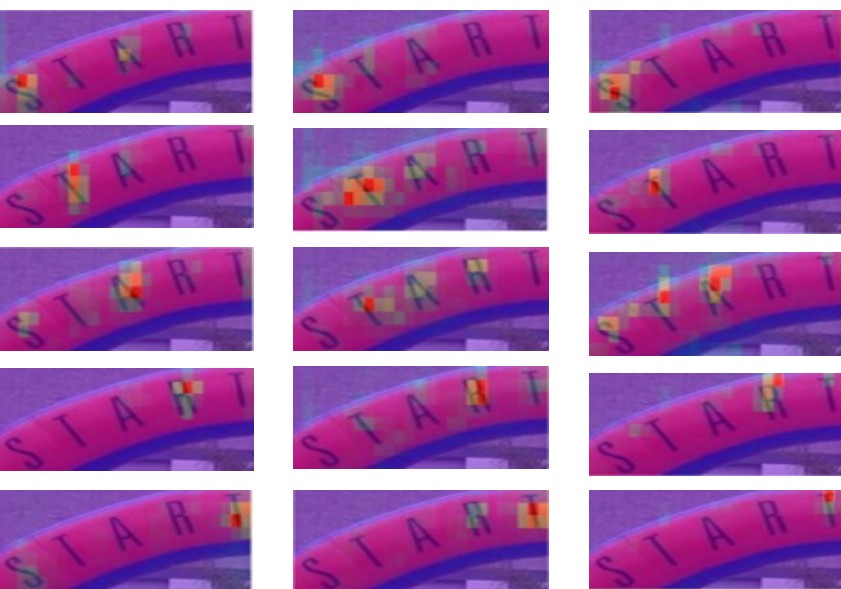

Figure 6: Quality plot of the attention map from the encoder output. Each row represents the attention of each character. **1-st** column shows the attention from CTCLoss. **2-nd** column shows the attention for the cross entropy loss. **3-rd** column shows the attention for the baseline method.

