# OpenReview forum: "NAText: Faster Scene Text Recognition with Non Autoregressive Transformer"
_ICLR.cc/2025/Conference — Submitted to ICLR 2025_

### Official Review · Reviewer_LM8b · 2024-10-29

**Soundness:** 2
**Presentation:** 2
**Contribution:** 2
**Rating:** 5
**Confidence:** 3

**Summary:**

This paper proposes a new non-autoregressive text recognition model, NAText, which aims to improve the speed and performance of the scene text recognition task.NAText is based on the Transformer architecture and introduces both progressive sampled learning and positional coding improvements to address the performance degradation of the non-autoregressive model. NAText is based on the Transformer architecture. Experiments show that NAText helps to better utilize the positional information for 2D feature aggregation. With all these proposed techniques, the NAText has achieved competitive performance to the state-of-the-art methods.

**Strengths:**

1) NAText's new use of positional coding allows the model to better identify the center of the character, which plays a positive role in improving the accuracy of text recognition.
2) The progressive sampling learning proposed in the paper helps the model to focus on learning difficult samples and increases the identification accuracy of non-autoregressive methods.
3) The paper was tested on multiple benchmark datasets and compared to multiple methods demonstrating strong advantages.

**Weaknesses:**

1) Lack of comparison with recent existing SOTA methods in Table 1, e.g., PIMNet, PARSeq.
2) Lack of evaluation of model performance in actual real-world scenarios.
3) No mention of the ability to generalize to other languages.

**Questions:**

1) Can you give a clearer and more understandable structural diagram of the encoder design?
2) Does the model's performance suffer in scenes with complex backgrounds (e.g., street scenes or billboards) or with heavily distorted characters?

---

### Official Review · Reviewer_FZsn · 2024-10-31

**Soundness:** 3
**Presentation:** 2
**Contribution:** 2
**Rating:** 6
**Confidence:** 4

**Summary:**

The paper introduces NAText, a non-autoregressive transformer model aimed at faster scene text recognition without sacrificing accuracy. The traditional autoregressive methods, while effective, are slow due to their sequential decoding. Non-autoregressive methods are faster but typically show degraded performance due to the assumption of character independence. To address this, NAText employs progressive sampled learning to focus on hard-to-predict characters and enhances positional encoding to capture character relationships more effectively. The paper demonstrates that NAText achieves competitive results compared to autoregressive and state-of-the-art models across several benchmark datasets, while being significantly faster.

**Strengths:**

1. NAText introduces an innovative progressive learning strategy that effectively mitigates the performance drop typically associated with non-autoregressive models, especially in handling irregular text and longer sequences.
2. The model strikes an excellent balance between speed and accuracy, achieving competitive performance with autoregressive models while operating nearly three times faster. This is crucial for real-time applications of scene text recognition.
3. The experimental section is thorough, comparing NAText across popular benchmarks and analyzing performance across different text lengths and types (regular vs. irregular). The detailed ablation studies further strengthen the paper by providing insights into the contributions of each component of the model.

**Weaknesses:**

1. In your training process, the initial predictions from your encoder are supervised, which means after many epochs of training, the encoder's predictions will become very close to the target, with only a few errors. In such a case, the decoder may only imitate the encoder's output and fail to learn anything. Did you encounter this issue, and how do you ensure that both the encoder and decoder are learning effectively?
2. As mentioned in the first point, your method calibrates the predictions after the initial prediction. However, the paper lacks the experimental results that show the difference between the initial predictions and the final calibrated predictions. If you can demonstrate that the encoder’s predictions had several errors but the decoder corrected them, it would prove the effectiveness of your two-stage method. Otherwise, your two-stage method may not fundamentally differ from a conventional end-to-end approach.
3. Your masking strategy is innovative but somewhat confusing to me. In previous works, masking is used to hide valid information to add challenges to the model’s predictions. However, your approach replaces the masked character with the ground truth, which, as I understand it, seems to merely reduce the learning difficulty in the early stages of training. The paper does not explain in detail why this method helps the model learn the relationships between characters, despite making such a claim.
4. A small question: if the encoder's predicted sequence length differs from the ground truth (which happens often), how do you align them when calculating the cross-entropy loss for the decoder’s predictions?

**Questions:**

Please refer to my previous comments.

---

### Official Review · Reviewer_ZCcQ · 2024-11-02

**Soundness:** 2
**Presentation:** 2
**Contribution:** 2
**Rating:** 3
**Confidence:** 4

**Summary:**

In this paper, the authors propose a simple yet effective non-autoregressive text recognizer, NAText. It addresses the issue of non-autoregressive models typically underperforming compared to autoregressive models. Specifically, the paper corrects the fundamental assumption of independence between characters in non-autoregressive decoding and designs progressive sampling learning to enhance the performance of non-autoregressive models. Additionally, the authors introduce position encodings with clear physical significance to achieve better visual perception. The effectiveness of the proposed method is validated on both regular and irregular datasets.

**Strengths:**

1）This paper introduces NAText as a simple yet powerful non-autoregressive scene text recognizer. Compared to recent works, it is both fast and robust.
2) The paper thoroughly investigates the reasons behind the poor performance of non-autoregressive decoding and proposes progressive sampling learning to overcome these issues.
3) By redesigning the decoder structure to leverage positional information, the paper achieves improved visual perception.

**Weaknesses:**

1）The progressive sampling proposed in this paper is, in strict terms, still a form of masking technique. It utilizes the self-attention interaction between character embeddings of predicted and unpredicted characters, which is not novel in the context of scene text recognition.
2) Regarding Section 3.2, "Decoupled Non-Autoregressive Decoder," there seems to be an issue with the attention formula in equation (3). It should be $Attention(q,k,v)=softmax(\frac{qk^T}{\sqrt{d_k}})v$. Additionally, the authors mention that $qk^T$ can be expressed as $c_{q}^{T}X + p_{q}^{T}X_p$. However, what purpose does the multiplication of the two position matrices serve, and what is the significance of performing cross-attention between them?
3) Although the authors describe their model as relatively simple, the model structure and training procedures suggest otherwise. It involves a two-stage training process, introduces additional character positional information, and requires prediction outputs at each decoder layer. This approach incorporates a considerable amount of extra information, such as character positional data and a pre-trained CTC decoder for initializing inputs to the non-autoregressive decoder. From this perspective, the proposed method does not seem to be a straightforward model, despite achieving significant performance improvements over other non-autoregressive approaches.
4) While the authors compare their method with numerous text recognition techniques in Table 1, there is a lack of comparison with more recent text recognition technologies. It is unclear when the authors wrote this paper, but the latest method they compare against appears to be a paper published in 2022, making such experimental comparisons less convincing. Moreover, the authors only conducted experiments on some existing datasets, without addressing more challenging datasets like Union14M-Benchmark in this paper.

**Questions:**

Please refer to the weaknesses part.

---

### Official Review · Reviewer_G887 · 2024-11-04

**Soundness:** 2
**Presentation:** 1
**Contribution:** 2
**Rating:** 1
**Confidence:** 4

**Summary:**

The paper presents a non-autoregressive (NAR) approach to scene text recognition, designed to achieve faster inference speeds by removing the iterative generation process required by autoregressive (AR) models. While NAR models typically suffer from reduced accuracy compared to AR models, this paper addresses these limitations through two key innovations:

1. Progressive Sampled Learning: A novel training strategy that progressively focuses on harder characters, helping the model capture character dependencies without iterative decoding.

2. Two-Stage Decoding: A refined two-stage decoding process, where an initial coarse prediction is used to guide a final refined prediction, improving robustness and accuracy.

With these improvements, the proposed method shows enhanced accuracy over previous NAR methods and narrows the performance gap with AR models.

**Strengths:**

1. The proposed method achieves competitive accuracy, outperforming previous non-autoregressive (NAR) methods like PARSeqN and approaching the accuracy levels of autoregressive (AR) models.

2. The method benefits from high inference speed due to its non-autoregressive design, making it efficient for real-time applications.

**Weaknesses:**

The paper presents some interesting ideas, and the results appear competitive. However, the presentation requires significant improvement. In its current form, the paper is barely readable, and certain sections require educated guesses to fully comprehend.

For example, Section 3 does not clearly describe the proposed method. Figures 1 and 2 lack explanations for the notations used, leaving readers to infer meanings. Additionally, many mathematical symbols are used without clear definitions—such as the domain for \( x \) and \( y \) in Equation 1, or the exact definition of \( PE(x, y) \). Furthermore, some equations lack rigor and seem arbitrary in notation; for instance, in Equation 5, the term \( \log p(y_t | y_{<t}, X; \theta) \) should use a more precise notation, such as \( y_1, y_2, \ldots, y_{t-1} \), and \( y \) itself should be defined.

**Questions:**

N/A

---

### Official Review · Reviewer_Fbyw · 2024-11-04

**Soundness:** 3
**Presentation:** 3
**Contribution:** 3
**Rating:** 5
**Confidence:** 5

**Summary:**

This paper proposes a non-autoregressive transformer-based framework to recognize scene text. It introduces a progressive learning strategy to learn the relationship between characters for hard cases. Moreover, the authors adopt the positional encoding of the character center to enrich the query representations.  Experimental results show the effectiveness of each core contribution and the excellent inference speed with competitive recognition accuracy.

**Strengths:**

+ This work focuses on improving the runtime with competitive performance. The inference speed is very important, and previous works usually ignore the speed.
+ The authors deeply analyze the reasons behind the inferior performance of non-autoregressive decoding. To deal with this issue, the authors adopt a progressive sampled learning strategy to capture the relationship between characters for hard cases.
+ The authors design the decoder structure by introducing the positional information to learn better visual feature representations, which helps improve the performance of scene text recognition.

**Weaknesses:**

- The authors will use the center coordinates of characters to encode the position information, so it seems that the character-level annotations will be involved. It may burden the annotations of the ground truth.
- The novelty of this paper is somewhat incremental. It seems that the proposed method introduces a progressive learning strategy and embeds the position of the character center into queries based on a non-autoregressive paradigm. In effect, some curriculum learning from easy examples to hard examples is explored, while some hard example learning strategies are also proposed.  I think the authors should illustrate the advantage of the proposed progressive learning strategy compared with similar learning strategies on hard examples.
 - It is a common operation that adds character-level position information to the queries. The authors should illustrate the difference between the re-designed decoder structure and the existing decoder networks.  It seems that the authors adopt the core techniques of the decoder in the scene text spotting to the scene text recognition.
 - The authors do not report the parameters of the proposed network.
 - In Table 1, the authors only compared the methods before 2022. Some latest works are missing.

**Questions:**

1. Whether do the authors adopt the character-level annotations to train the model?
2. How do the authors define the hard cases?
3. What are the advantages of the proposed progressive learning strategy compared to existing hard example learning strategy on other tasks? Can the hard example learning strategy be directly applied to the scene text recognition task?

---

### Official Review · Reviewer_AYCT · 2024-11-04

**Soundness:** 2
**Presentation:** 2
**Contribution:** 2
**Rating:** 5
**Confidence:** 5

**Summary:**

This paper addresses the challenge of improving efficiency in scene text recognition by leveraging a non-autoregressive Transformer model. Traditional autoregressive methods, while accurate, are slow due to their sequential decoding process. In contrast, non-autoregressive methods allow parallel decoding, significantly increasing speed, though often at the cost of accuracy.

**Strengths:**

1. This paper demonstrates originality by reimagining the structure and training of non-autoregressive models for scene text recognition, a field traditionally dominated by autoregressive approaches.
2. The authors present the content clearly, with well-defined objectives and a logical flow from motivation to implementation and results.
3. This work is significant as it addresses a fundamental trade-off in scene text recognition—accuracy versus speed—that has hindered the broader application of such models in real-time or resource-constrained settings.

**Weaknesses:**

1. Limited Exploration of Trade-offs in Progressive Sampling: The current ablation study outlines improvements from progressive sampling, but it lacks details on how varying the sampling strategy impacts training convergence time, model complexity, and potential overfitting on challenging characters.
2. Insufficient Comparison with Broader State-of-the-Art Non-Autoregressive Techniques: Although the paper compares NAText with several autoregressive and non-autoregressive models, the non-autoregressive comparison group could be expanded to include more recent advancements in non-autoregressive Transformers and efficient decoding strategies beyond the baseline models presented.
3. Ambiguity in Efficiency Gains for Real-World Applications: While the model demonstrates significant speed improvements in experimental settings, it is unclear how these gains translate to various real-world applications, particularly in resource-constrained or edge computing environments.
4. Positional Query Design Clarification: Although the paper introduces a redesigned positional query with character center encoding to enhance 2D feature aggregation, the impact and implementation details of this component could be clarified.
5. ack of Qualitative Analysis and Error Case Discussion: The paper would benefit from including qualitative examples or visualizations of both successful and challenging cases for NAText.

**Questions:**

1. Could you clarify if there were any observed trade-offs with the progressive sampled learning technique, such as increased training time or risk of overfitting on challenging characters?
2. Can you provide further comparisons with recent non-autoregressive Transformer-based methods, such as those used in non-autoregressive machine translation or other sequence tasks?
3. The paper reports notable speed gains, but could you share more specific efficiency metrics, such as memory usage or latency on resource-limited devices like mobile GPUs?
4. The positional query design is an interesting addition, but its computational impact remains unclear. Does this design increase complexity in a meaningful way, and if so, how does it balance this with performance gains?
5. Could you provide qualitative examples or visualizations of cases where NAText performs well and where it struggles, especially with irregular or noisy text?
6. Since autoregressive models often struggle with longer sequences, did you conduct experiments or observe particular trends with NAText’s performance on varying text lengths?

---

### Official Review · Reviewer_cVyZ · 2024-11-05

**Soundness:** 2
**Presentation:** 3
**Contribution:** 2
**Rating:** 5
**Confidence:** 5

**Summary:**

The paper presents NAText, a straightforward and effective non-autoregressive text recognizer that aims to close the performance gap with autoregressive models. The authors tackle this challenge by redefining core assumptions and introducing progressive sampling, which helps the model handle difficult cases more effectively. They also add a physically interpretable positional encoding, enhancing the model’s ability to capture visual information.

**Strengths:**

- The motivation is good.
- The method achieves a good balance between accuracy and inference time.
- The two-stage prediction design is compelling, beginning with a coarse prediction that is subsequently refined for greater precision.

**Weaknesses:**

- The related works compared in this study are all from before 2022, lacking comparisons with the latest research, such as:

      [1] OTE: Exploring Accurate Scene Text Recognition Using One Token, CVPR 2024
      [2]  Multi-modal In-Context Learning Makes an Ego-evolving Scene Text Recognizer, CVPR 2024
      [3] Bridging the Gap Between End-to-End and Two-Step Text Spotting, CVPR 2024
      [4] SPTS v2: Single-Point Scene Text Spotting,TPAMI 2023

- This approach requires character and character position annotations, which are often unavailable in real-world data. Especially for irregular text, precise character positioning is difficult to obtain and costly to annotate. Addressing this limitation would improve applicability.

- The datasets chosen for comparison are relatively simple, with high accuracy already achieved by many methods. To better validate the method’s effectiveness, more challenging datasets, such as Total-Text or Union14M [5], could be considered.
      [5] Revisiting Scene Text Recognition: A Data Perspective, ICCV 2023

**Questions:**

- How to get  the character’s center coordinate (x,y)?
- The model uses a two-stage testing process: it first extracts image features to generate a coarse sequence prediction, which is then passed to the decoder for a final refined result. This two-stage setup may be time-consuming, so it would be helpful if the authors clarified whether the reported inference time covers both stages or just the first. Additionally, providing the model’s total parameter count would aid in evaluating its efficiency and complexity.

---

### Meta-Review · Area_Chair_HbHd · 2024-12-20

**Metareview:**

# Summary of the Paper
**Key Contributions**:
1. **Non-Autoregressive Transformer**: Proposes NAText, a non-autoregressive (NAR) transformer designed for scene text recognition to improve inference speed over traditional autoregressive (AR) methods.
2. **Progressive Learning Strategy**: Introduces a training strategy that progressively focuses on harder characters to improve model accuracy.
3. **Enhanced Positional Encoding**: Redesigns the positional query using character center information to improve 2D feature aggregation.

**Experimental Results**:
- Demonstrates competitive accuracy compared to state-of-the-art AR models while achieving faster inference speeds.
- Tested on multiple datasets, showing good performance on regular and irregular text.

---

# Justification for Rejection

While the paper has strengths, it fails to address critical issues that limit its acceptability:

## 1. Incremental Contributions
- **Progressive Learning Strategy**: While novel for the task, it closely resembles existing hard-sample learning strategies. The authors do not sufficiently differentiate their approach from prior work, such as curriculum learning or other masking techniques.
- **Positional Encoding**: The use of positional encodings is well-established in related domains. The added contribution of using character center coordinates lacks novelty and broader implications.

## 2. Experimental Limitations
- **Dataset Choices**: Comparisons are limited to older benchmarks, omitting recent and more challenging datasets like Union14M. This makes the experimental evaluation less convincing.
- **Comparisons with Recent Work**: The study primarily compares with methods published before 2022, ignoring advancements in scene text recognition from 2023–2024, such as PIMNet or PARSeq. This undermines the relevance of the results.
- **Performance in Real-World Scenarios**: There is no evaluation on real-world datasets with complex backgrounds or multilingual settings, which are critical for practical applications.

## 3. Methodological Weaknesses
- **Dependency on Character-Level Annotations**: The proposed method relies heavily on character-level annotations, which are costly and often unavailable in real-world scenarios. This limits its scalability and applicability.
- **Two-Stage Decoding**: Although faster than AR models, the two-stage decoding may introduce additional computational complexity compared to simpler NAR approaches. The paper lacks clarity on the efficiency trade-offs.

## 4. Presentation Issues
- **Clarity and Rigor**: The methodology section is poorly explained, with insufficient details on key components such as the progressive sampling strategy and positional query design. Figures and equations lack proper annotations, making the paper difficult to follow.
- **Lack of Qualitative Analysis**: The absence of qualitative examples or error analysis reduces the interpretability and practical insights of the proposed method.

## 5. Limited Impact
- While the paper achieves speed improvements, the accuracy remains below state-of-the-art AR models. The results do not convincingly demonstrate that the performance gap between AR and NAR methods has been meaningfully closed.

---

# Recommendation

Based on the above considerations, I recommend **rejection**. The paper makes incremental contributions that are insufficiently validated, lacks comparisons with recent methods and datasets, and suffers from significant presentation issues. While the focus on inference speed is important, the overall novelty and impact of the work fall short of the standards for a high-impact venue. Further revisions with more robust experiments, clearer methodology, and expanded comparisons are necessary for reconsideration.

**Additional Comments On Reviewer Discussion:**

The authors did not provide rebuttal.

---

### Decision · Program_Chairs · 2025-01-22

Reject